# Multidimensional Data Interpretation of Vibration Signals Registered in Different Locations for System Condition Monitoring of a Three-Stage Gear Transmission Operating under Difficult Conditions

**DOI:** 10.3390/s21237808

**Published:** 2021-11-24

**Authors:** Grzegorz Wojnar, Rafał Burdzik, Andrzej N. Wieczorek, Łukasz Konieczny

**Affiliations:** 1Department of Road Transport, Faculty of Transport and Aviation Engineering, Silesian University of Technology, 40-019 Katowice, Poland; Rafal.Burdzik@polsl.pl (R.B.); lukasz.konieczny@polsl.pl (Ł.K.); 2Department of Mining Mechanization and Robotisation, Faculty of Mining, Safety Engineering and Industrial Automation, Silesian University of Technology, 44-100 Gliwice, Poland; Andrzej.N.Wieczorek@polsl.pl

**Keywords:** vibrations, gear transmission, sensor location, STFT, FFT, waveform

## Abstract

This article provides a discussion of the results of studies on the original system condition monitoring of a three-stage transmission with a bevel–cylindrical–planetary configuration installed in an experimental scraper conveyor. Due to the high vibroactivity of gear transmissions operating under the impact of a scraper conveyor’s chain drive, these unwanted effects of machine operating vibrations were assumed to be applied. For purposes of the study, vibrations were measured on the driving transmission housing in an idling scraper conveyor. The main purpose of the study was to establish the frequencies characteristic of the gear transmission, and to determine whether it was possible to run vibroacoustic diagnostics of the same transmission under conditions with a considerable impact of the conveyor chain. An additional cognitively significant research goal was the analysis of the dependence of the diagnostic utility of the signal depending on the sensor mounting point. Five different locations of three-axis sensors oriented to the next stages and various types of gears were determined, as well as places characterized by high spatial accessibility, which are often selected as places for measuring the vibration of gears. Using MATLAB software, a program was written that was calibrated and adapted to the specifics of the measuring equipment based on the collected test results. As a result, it was possible to obtain a multidimensional data interpretation of vibration signals of system condition monitoring of a three-stage gear transmission operating under difficult conditions. The results were based on signals registered on the real three-stage gear transmission operating under the impact of a scraper conveyor’s chain drive.

## 1. Introduction

What proves to be essential in the overall practice of diagnostics is vibroacoustic diagnostics, which make it possible to preliminarily assess the manner in which a defect develops in a relatively simple and cheap way. The most fundamental goal of operational diagnostics is to detect defects at early stages of the related changes, before a transmission becomes actually damaged. An assessment of the dynamic state of a transmission based on an analysis of vibroacoustic signals is a problem discussed in numerous papers [1,2,3,4]. It should be noted that there are applicable standards and recommendations that primarily consider the aspects of fatigue strength, while disregarding the nature of the transmission damage at the same time. Under industrial conditions, researchers typically measure the root-mean-square value of velocity or acceleration of the transmission housing vibrations [5]. What must be emphasised is that the vibration level value is significantly dependent on the measuring conditions, as well as the changes to the value of an external load. Under variable load conditions, it is all the more necessary to normalise the diagnostic signal alternatively to apply diagnostic measures characterised by low sensitivity to changes in the operating conditions of the transmission. Furthermore, when dealing with a driving transmission of a heavy-duty scraper conveyor operating under the difficult conditions of an underground mine heading, there are significant constraints that must be taken into account in terms of the possibility of using a diagnostic system. This article provides a discussion on the results of studies of vibroactivity of selected components of gear transmissions driving the power transmission system in a scraper conveyor.

In a typical reliability serial structure, which a scraper conveyor represents, there is a need to ensure high operational reliability of all its elements, because the elimination of one of the elements forces the entire machine system to stop. Driving elements of transport systems; i.e., gears and bearings of drive wheels, usually work in an condition of high aggressiveness, which consists of the impact of groundwater (often highly saline), aggressive gases and exhaust gases of diesel engines containing sulfur and nitrogen oxides, and other aggressive chemical compounds. The presence of erosive agents such as stone dust, coal dust, pyrites, and possibly quartz sand residues, is also significant. It significantly increases the tribological wear of all cooperating elements of the drive system [6].

The progressive wear of the driving elements causes the deterioration of the operating conditions of the transmission, an increase in the dynamic surplus in the tendons, the deterioration of the mechanical efficiency, and a significant increase of the process of operational destruction of the elements of machine systems. During operation, there are also additional unfavorable destructive factors: electrochemical corrosion and dynamic shocks related to system overload. These factors are among the main causes of failures leading to downtime of operating systems and accidents of employees, negatively affecting the level of work safety. It should be added that as a result of the joint interaction of the aforementioned destructive factors, a synergy of their impact occurs, which results in the intensification of destructive processes exceeding the simple addition of intensification of individual types of destructive processes [7].

One method of counteracting destructive processes is the use of materials with increased resistance to abrasive wear and the common oversizing of the structure of drive units, in particular the size of bearings, thickness of supporting sheets, and dimensions of gear wheels. The applied solution, on the one hand, does not ensure the achievement of appropriate service life due to the synergistic nature of the impact of destructive processes; on the other hand, it leads to irrational management of raw materials and energy, which leads to the unsustainable development of the industry [8].

In the current state of the art, there have been attempts to use technical condition monitoring systems, but the introduction of diagnostic elements—sensors and signal cables—attached to the outside of the body of power units makes them vulnerable to damage. In addition, the sensoring of drive units in places easily accessible for maintenance services does not ensure obtaining a signal containing full information about the condition of these machines. At this point, it should be added that the drive units of scraper conveyors are often located in places with difficult access for service personnel, and often also are covered with a layer of spoil. In underground mining, there is a technical infrastructure that allows the implementation of diagnostic systems; in this respect, we should mention the dispatching systems supervising the haulage processes, monitoring and controlling the operation of conveyors, as well as the presentation of data from process parameter sensors [9,10].

In addition to all these factors that significantly affect machine vibrations, and thus affect the diagnostic information contained in the signal, the correct selection of the location of the measuring sensors is also important. Therefore, an additional aim of this research was to analyze the influence of the sensor location on the content of information about the technical condition and operating parameters in vibration signals.

The problem of diagnosing drive transmission systems, including those equipped with toothed gears, is a difficult scientific topic [11], especially when diagnostics are carried out on the basis of vibration signals. For this reason, the recorded signals are subjected by scientists to many types of analyses; e.g., in the time domain (synchronous averaging [12] or signal resampling [13]), in the frequency domain [14], and in the time–frequency domain [12,15,16], including the very often used short-time Fourier transform (STFT) [13,15,17,18,19,20,21]. In addition, signal analysis methods using artificial neural networks and genetic algorithms are used [16,22]. The methods of detecting various faults of gears are improved both on the basis of signals recorded on real objects and also obtained on the basis of simulation tests [1,15,22,23] based on dynamic models. In some papers, the authors also dealt with the problem of detecting damage to gears in the case of simultaneous detection of bearing failures [24]. Based on the analysis of the literature, it can be concluded that the best results in detecting the damage of gear elements are obtained if the tests are carried out in the case of relatively simple gears; e.g., a single-stage spur gear [25,26]. For this reason, however, the authors of this paper undertook the subject of signal analysis, also using the popular as-shown short-time Fourier transform (STFT) in the case of a more complex object; i.e., the drive system of a mining scraper conveyor equipped with a three-stage bevel–cylindrical–planetary gear transmission and cooperating with the chain drive, generating additional random vibration excitation. The results of such tests are important because often in publications, scientists in a single experiment present the possibility of detecting one or several failures, but a much more difficult topic is to detect these failures during normal machine operation, when reporting false alarms or missing alarms generates significant costs, as presented in [27].

## 2. Object of Study

The object of the study was a type KPL-25 transmission with a ratio of I = 39 and a power of 400 kW. Figure 1 depicts this transmission and its dimensions. Table 1 summarises the transmission’s basic kinematic and load parameters.

The gear transmission subjected to study was enclosed in a housing composed of grey cast iron, and was a three-stage type transmission, comprising:-1st stage—bevel gear;-2nd stage—cylindrical gear;-3rd stage—planetary gear.

The cylindrical gears of the transmission studied were made of alloy steel subject to heat treatment, thermochemical treatment, and grinding. The bevel gears featured curved teeth of a cyclo-poloidal geometry, the cylindrical ones had skewed teeth, and the planetary stage had straight toothing. The teeth of all gears were manufactured to the 6th class of precision as per PN-ISO 1328.

The gear transmission in question was installed in an experimental scraper conveyor (Figure 2). Its 100 kW driving motor operated at 1470 rpm and was coupled to the transmission via a flexible bush coupling, while a splined coupling provided its connection to the transmission shaft.

## 3. Research Design and Sensor Placements

The main goal of this study was to obtain a multidimensional data interpretation of vibration signals of system condition monitoring of a bevel gear transmission operating under difficult conditions. The secondary goal was to obtain a preliminary assessment concerning the possibility of using vibration signals in the diagnostics and evaluation of the technical condition of the power transmission system in a scraper conveyor’s driving transmission. The vibroacoustic signals recorded during tests and the signal measures based on the former should reflect individual symptoms demonstrating how intense and advanced the wear is. A phenomenon encountered in gear transmissions is modulation of the carrier signal, which typically corresponds to the high-frequency signals of meshing or resonant frequencies of the transmission’s components. Amplitude and frequency modulation in a gear transmission may be observed due to various reasons that have been described in detail in the literature on the subject. 

The basis of a correct diagnostic process is the identification of the technical condition of the machine. This is also the primary task of a health monitoring system (HMS). In the case of such complex machines operating in difficult conditions in complex mechanical systems, there is often a phenomenon of overlapping symptoms of the condition of individual machine elements and disorganization of components resulting from damage or wear. These unfavorable phenomena are intensified in the case of vibration signals, which carry all dynamic information of the machine, so the separation and classification of symptoms on the basis of vibration measures is a very big challenge.

One of the preferred approaches to this issue is the proper selection of the location of vibration sensors, in accordance with the principle “as close to the source of damage as possible”. Due to the difficulties in mounting sensors in such places, and often the lack of knowledge about potential sources of damage or too many such threats, this approach has large application limitations. In addition, the selection of a site for a contoured source of damage may make it difficult to identify other threats.

Due to such a difficult research issue, an active diagnostic experiment was planned, the aim of which was to compare the identification quality of the technical condition of the gear transmission depending on the mounting location of the vibration sensor.

The signals were recorded with a sampling frequency of 50 kHz, which covered the entire range of analysis. A sampling frequency of 50 kHz far exceeded the bandwidth of the analyses presented in this article. This article, however, presents only a part of a larger study, which also included measurements of sound pressure, for which the analysis bandwidth covers 21 kHz; therefore, according to the Nyquist–Shannon theorem, signals should be sampled with a frequency at least two times higher. One of the assumptions of complex vibroacoustics research is to analyse all signals (vibration and acoustics) with synchronisation. The same time vector allows the comparison of the correlation between vibration and acoustic signal symptoms. Therefore, it was decided to use 50 kHz. Of course, the appropriate frequency band according to the type of sensor used and the type of fixation used only covered the band up to 5 kHz, which is presented in the article.

The scheme of the measurement system is depicted in Figure 3. The acceleration of vibration was measured by three axial PCB sensors (model 356A02 ICP). Piezoelectric accelerometers are widely used for measuring acceleration. Part of their appeal is a flat frequency response, if the correct accelerometer and mounting method are chosen. For the data acquisition, an Ni 9233 analog–digital device was used. General parameters of the measurement system are depicted in Table 2.

The actual measurements were conducted in an unloaded idling transmission, during an operating phase commonly referred to as running-in. The measurements included recording vibration signals of the gear housing. The sensor locations are depicted in Figure 4. Sensor 1 was mounted on the housing close to the 1st stage transmission—bevel gear. Sensors 2 and 3 were mounted on the housing close to the 2nd stage transmission—both cylindrical gears. Sensor 4 was mounted on the housing close to the 3rd stage transmission—planetary gear. Each of the gears represents different stages of the transmission and performed different vibroactivity. Sensor 5 was mounted on the top of the housing.

All applied sensors had the same specifications, including sensitivity, and all of them were mounted the same on the housing by means of a magnetic base. Sensor mounting can significantly affect the overall vibration and its spectral data. The most important factor is the mounted resonance as the resulting change in natural frequency, caused by the structural change of the accelerometer, based on the mounting method used. This change in natural frequency is a direct result of the change in mass and stiffness. Sensors must be coupled so that complete event information is transferred. Mounting methods may vary, with some transferring event information more effectively than others. The given usable bandwidth may be reduced by the mounting method. Therefore, the method of attaching the accelerometer to the measuring surface is one of the most critical factors in obtaining accurate results at high frequencies for practical vibration measurements. When a flat magnet is used, analysts should expect to yield good data, up to roughly 7000–8000 Hz before reaching 3 dB and entering the amplification range. However, with perfect installation and execution, some analysts report frequency responses as high as 10,000 Hz. 

In our case, the observed bandwidth was up to 5 kHz, thus a magnetic base could be used. During research, the vibration accelerations were registered using a three-axis piezoelectric transducer. The research design is schematically depicted in Figure 5.

The issues related to the condition monitoring systems of multistage gear transmissions operating under difficult conditions and all the problems related to the diagnosis of these devices have resulted in new methods of recording and processing diagnostic signals constantly being sought. Alternative techniques for vibration measurements are current and torque measurements. They require intervention in the design of the devices, and have great limitations in the scope of possible defects to be detected. Other alternative techniques are based on thermal or noise measurements. In these cases, the diagnostic symptom detection algorithms are even more complicated; in addition, the working environment of mining gear actually makes it impossible to effectively conduct such measurements in isolation from external disturbances.

## 4. Method of Analysis

As a result of the studies, the transmission’s vibration signals were recorded. Digital signals were analysed using MATLAB software. For each sensor placement point, a matrix with a size of 3 × 1,600,000 was obtained, and additionally, a matrix consisting of 3 sampling time index vectors with the same dimension was determined. Such a set of values was each time a set of measurement data. For multidimensional data interpretation of vibration signals of the bevel gear transmission, three stages of analysis were conducted.

Vibration analysis begins with a time-varying, real-world signal from a transducer or sensor. The first stage of the analysis was the waveform of signals. At this stage, the energy content of the signal, the distribution of the maximum values in the signal waveform, and basic statistical values as estimates of representative values of the signal were estimated. An important diagnostic feature of vibration signals are also measures of dissipation and dispersion. As quantity estimators of waveform of n-samples, signals were calculated as follows:(1)$$\bar{x}=\frac{1}{n}\left(\overset{n}{\underset{i=1}{\sum }}x_{i}\right)$$
(2)$$RMS_{x}=\sqrt{\frac{1}{n}\left(\overset{n}{\underset{i=1}{\sum }}{\left|x_{i}\right|}^{2}\right)}$$
(3)$$std_{x}={\left(\frac{1}{n}\overset{n}{\underset{i=1}{\sum }}{\left(x_{i}-\bar{x}\right)}^{2}\right)}^{\frac{1}{2}}$$
(4)$$CoV_{x}=\frac{RMS_{x}}{std_{x}}$$

The root-mean-square value (*RMS_x_*) is the most useful, because it is directly related to the energy content of the vibration profile. It relates to the power of the wave. The root-mean-square value is one of the important factors for machinery condition monitoring. The standard deviation (*std_x_*) is a measure of the variability of a signal about its mean value ($\bar{x}$). The standard deviation is invariant under changes in location, and scales directly with the scale of the random variable. The mean and standard deviation of a set of data are descriptive statistics usually reported together. In a certain sense, the standard deviation is a “natural” measure of statistical dispersion if the center of the data is measured about the mean. For a vibration signal with a mean value of zero, the standard deviation is equal to the root-mean-square value of the signal. In probability theory and statistics, the coefficient of variation (*CoV_x_*) is a standardized measure of the dispersion of a probability distribution. The coefficient of variation is useful because the standard deviation of data must always be understood in the context of the mean of the data [28,29]. It is widely used to express the precision and repeatability of an assay. Therefore, it can also be a diagnostic measure, assuming that the symptoms of successive failures increase the vibration signal dispersion. A sample time-varying vibration signal with calculated quantity estimators has been depicted in Figure 6.

The second stage of analysis is focused on the identification of dynamical components in the signal. To enhance the feature extraction capability, the time-domain data is proceeded to generate frequency-domain data. The Fourier transform can be calculated by using the fast Fourier transform algorithm [30,31,32]. The algorithm to compute discrete Fourier transforms (DFTs) is the fast Fourier transform (FFT) [33], which can reduce the computational complexity of discrete Fourier transform significantly [34]. The DFT, its most general form, is inefficient for machine computation, requiring *N*^2^ complex operations for a signal containing *N* samples. This motivates the development of the fast Fourier transforms, a family of efficient implementations of the DFT for different composites of *N*. In particular, we examined the radix-2 Cooley–Tukey FFT algorithm [35], which reduces the number of complex operations required for *Nlog*_2_(*N*) [36]. The most commonly used FFT algorithm is named after J.W. Cooley, an employee of IBM, and J.W. Tukey, a statistician, who jointly developed an implementation of the FFT for high-speed computers in 1965 [37,38,39].

The discrete Fourier transform is an approximation of the continuous Fourier transform for the case of discrete functions. Given a real sequence of {*x_n_*}, the DFT expresses them as a sequence {*X_k_*} of complex numbers, representing the amplitude and phase of different sinusoidal components of the input signal [36].

Signal processing was conducted in MATLAB. For the calculation, discrete Fourier transform (DFT) of the vector was computed with a fast Fourier transform (FFT) algorithm. The fast Fourier transform algorithm treated the columns of a matrix as vectors, and returned the Fourier transform vector for each column, leading to a Fourier transform matrix. The discrete Fourier transform (DFT) of a signal x may be defined by:(5)$$X_{k}=\overset{N}{\underset{j=1}{\sum }}x\left(j\right)\omega_{N}^{\left(j-1\right)\left(k-1\right)}$$
(6)$$\omega_{N}=e^{\left(-2\pi i\right)/N}$$
where $\omega_{N}$ is an *N*th root of unity.

The transformation of the signal from the time domain to the frequency domain based on the Fourier series and the FFT transform enabled the identification of components related to the dynamics of the machine operation. This allowed for the identification of parameters of kinematic nodes and other dynamic phenomena, including changes in machine parameters resulting from lab damage [40]. Therefore, this stage of the analysis of the results was essential. Examples of results and conclusions resulting from their analysis are presented in Figure 7.

As shown in Figure 7, the frequency analysis of the FFT spectra enabled the identification of components correlated with the kinematic parameters of the transmission (Figure 7a), which was the basis for condition monitoring. However, in the spectra, there may also have been other signal components (Figure 7b) that were not correlated with the machine operation parameters, and may have been damage symptoms. In the case of overlapping of these components in the spectra, the diagnostic result may have been difficult or erroneous. In the case of such nonstationary signals, the methods of observation in time and frequency domains should be used simultaneously. Moreover, noise generated by devices and machines can be a significant issue for their users. For acoustic signals, prolonged exposure to high-level noise, as in some industrial environments, can lead to hearing damage [41]. In our case, the fundamental bandwidth was lower than the sensor (sensors datasheet) and magnetic mounting properties (according to ISO 5348), and was less than 6 kH.

In our case, while the transmission was operated, it was subject to various wear and degradation processes that affected the parameters of the modulating signals. Adequate processing of the signals in the domains of time and frequency makes it possible to determine the extent and type of the carrier signal’s modulation. One of the methods enabling this is an analysis of a signal acquired in the domains of time and frequency simultaneously. 

The signals were also analysed in the domains of time and frequency. The short-time Fourier transform (STFT) is a simple and effective method widely used in machine diagnostics [20]. The STFT delivers a three-dimensional spectrum representing the behaviour of the signal amplitude in the time and frequency domain.

The following is the short-time Fourier transform equation:(7)$$S\left(b,f\right)=\overset{\infty }{\underset{-\infty }{\int }}x\left(t\right)\cdot e^{-j2\pi ft}\cdot w\left(t-bR\right)dt$$
where: (*t* − *b*)—window width;*w*(*t*)—Window function;*R*—Hop size between successive DFTs.

The hop size is the difference between the window length and the overlap length.

The STFT algorithm is used as follows. The frequency analysis is conducted by application of the fast Fourier transformation (FFT) to the following fragment of the signal, multiplied by the window function with the constant width of *w(t − b) = const*. The following fragments are analysed independently. The main disadvantage of this method is the constant window width. For example, when using a narrow window in the time domain, a good time resolution can be obtained, but the resolution in the frequency domain is worse. For a rectangular window, the jump function changes at the beginning, and the end of the windows are the source of a leak in the spectrum. Other windows are often used (e.g., triangular, Hanning, and Hamming) to minimise this effect [42]. Thus, the window’s width is considered as a compromise between the resolution in the time and the frequency domain. To improve the resolution in the frequency domain, the zero-complementing method can be used. This method consists of adding samples with a zero value of amplitude to the original signal to multiply the number of samples of the signal. Another method enabling the selectivity of the STFT method to be improved is the superposition of windows (each sample is used several times for a single FFT process). This paper presents some test results concerning vibration acceleration. 

The STFT with the Hamming window were used for signal processing. Each window of the signal was 100% elongated by using the complement zero method and analysed through the FFT process. The windows were superpositioned at 50% (Figure 8).

The modified STFT algorithm prepared in this way was the result of analytical experiments aimed at adjusting the transformation parameters of the recorded signals and the kinematic characteristics of the tested transmission gear.

An example of the STFT distribution of the vibration signal of the tested transmission gear obtained as a result of the above algorithm is shown in Figure 9.

The exemplary results of the STFT transformation of the signal as 3D and 2D distributions are shown in Figure 9. Due to the multidimensional analysis of the signal, the time distribution of individual frequency components was clearly visible, which was not always constant. Such information about changes in the amplitudes of specific frequency components over time may also be essential in SHM systems, especially in the diagnostic aspect of complex machines operating in heterogeneous and difficult conditions, such as mining conveyors.

To a large extent, diagnostics of toothed gears is based on signal-processing algorithms (usually vibration ones), the purpose of which is to separate signal components and identify damage symptoms. Recent trends in signal processing show new applications. The study in [43] presented a planetary gearbox fault diagnosis method based on continuous vibration separation and minimum entropy deconvolution. Continuous vibration separation was used to overcome the modulation effect caused by planetary movements, as well as restraining noise and asynchronous signal components, minimum entropy deconvolution was used to enhance fault-induced impulses if exist. Another approach was presented in [44]. Applications of novel decision trees for the classification of spectral-based 15D vectors of diagnostic data were developed. The study applied a combination of spectral analysis and the application of decision trees to a set of spectral features. Such multidimensionality of diagnostic data recognized the gearbox condition, even in nonstationary operating conditions. To enhance the performance of demodulation analysis for symptom separation, signal decomposition often is employed by using wavelet transform to decompose a signal with a clear modulation effect. Wavelet decomposition requires a physical understanding of the modulation effect to isolate the fault-related signals. The study in [45] presented a cepstrum-assisted empirical wavelet transform to solve this challenge. In the presented method, the vibration signal was decomposed using empirical wavelet filters designed based on the smoothed spectrum from the cepstrum analysis.

For SHM of gear transmissions, wear mechanism identification is an important issue. The authors of [46] presented a study in which wear evolution was tracked using an indicator of vibration cyclostationarity. More specifically, with consideration of the underlying physics of the gear-meshing process, and the unique surface features induced by fatigue pitting and abrasive wear, the correlation between the tribological features of the two wear phenomena and gear-mesh-modulated second-order cyclostationary properties of the vibration signal was investigated.

## 5. Results

The developed method of comprehensive analysis of vibration signals enabled full identification and selection of significant signal components in the amplitude, frequency, and time–frequency dimensions. The results were obtained using the signal-processing algorithm described in the previous section according to the research design diagram (Figure 5). It is particularly applicable to highly modulated signals, even at the stage of identifying the technical condition and operating conditions of the machine. The analyzed case concerned a three-stage transmission comprising a 1st stage—bevel gear, 2nd stage—cylindrical gear, and 3rd stage—planetary gear of a mining conveyor, which is an example of a machine consisting of many vibroactive elements cooperating with each other. 

The purpose of the developed method was a multidimensional comparison of the information content of signals registered at various points in the gear housing.

This paper presents the results obtained for two exemplary measuring points registered for the same operating conditions with a constant rotational speed of the motor supplying the transmission.

The first measurement point (marked as 1 in Figure 4) on the housing was close to the 1st-stage transmission—bevel gear near the point of the transmission connected to the driving motor (Figure 10).

The waveforms of vibrations in three orthogonal axes are depicted in Figure 11.

The spectra of vibrations in three orthogonal axes are depicted in Figure 12. 

As the last stage of analysis, STFT proceeded, and a time–frequency (t–f) representation of vibrations was established, as depicted in Figure 13. The time resolution was 0.16 s and the frequency resolution was 1.53 Hz.

The second measurement point (marked as 4 in Figure 4) on the housing was close to the 3rd-stage transmission—planetary gear near the point of the transmission connected to the to the scraper conveyor body (Figure 14).

The waveforms of vibrations in three orthogonal axes are depicted in Figure 15.

The spectra of vibrations in three orthogonal axes are depicted in Figure 16.

As the last stage of analysis, STFT proceeded, and a time–frequency (t–f) representation of vibration was established, as depicted in Figure 17.

To analyze and evaluate the information content in waveform vibration signals recorded at various measuring points, quantitative estimators were determined; these are summarised in Table 3. The estimators were selected to represent the energy content of the signal, the distribution of the maximum values, and measures of dissipation and dispersion. The basic parameters of the statistical analysis of the obtained values were also determined, such as mean value, standard deviation, range as the measure of the spread of extreme values, and variance.

The obtained results confirmed a large discrepancy in values depending on the measuring point, which, for the same machine operating parameters and technical condition, may result in an incorrect diagnostic process or incorrect identification of the machine operating condition.

The dispersion of the values of these parameters during the experiment on a real object is presented in the charts below (Figure 18). On the abscissa axis, a number of measurement points were marked, and for comparison, the designated statistical measures, such as mean value, standard deviations, and the measure of the range of extreme values are noted.

Then, the frequency distributions of vibrations as FFT spectra were subjected to comparative analysis. The values of the amplitudes of elementary frequency components were compared, as well as the appearing uncorrelated components, with the kinetic parameters of the transmission, which may ultimately have been symptoms of machine damage. Figure 19 shows an example of a comparison of the spectra of vibrations along the vertical axis (Z) determined for signals recorded at various measurement points, with the frequency bands of elementary components marked with dashed lines.

A strong correlation of the characteristic frequencies is clearly visible; however, significant differences can be observed when comparing the amplitude values of the dominant FFT components. Additionally, in the case of the spectra of the signals recorded at points 4 and 5, there was a much larger share of components in higher frequencies, where there were also local maxima. These bands reduced the amplitudes of the elementary frequencies and potentially indicated the occurrence of other phenomena not correlated with the kinematics of the gear, which may result from occurring failures.

Based on the analysis of the above graphs, an analysis and preliminary evaluation of noise in the recorded signals could be performed. The first observation showed clear quantitative differences in the noise occurrence, depending on the location of the measuring point (please compare the number of components above the frequency of 1500 Hz for the 1st and 2nd measurement points and the 4th and 5th measurement points). The first conclusion resulting from this analysis was the indication of the best measuring point for the identification of transmission operating parameters. On the other hand, when looking for symptoms of potential damage, especially in the initial stage of formation, information often appears in higher frequency bands as the effects of nonlinear phenomena. Therefore, if we want to develop a diagnostic measure that detects symptoms of damage at an early stage (diagnostic prediction), they are very often low-energy components occurring in higher frequency bands. In this case, it may have turned out that for diagnostic prediction purposes, measuring points 4 and 5 was more useful due to the content of high-frequency components. Of course, the greatest difficulty was in separating this useful information (symptoms) from the noise components and their effects. For this reason, we decided not to apply filtering of the signal to reduce noise in order to enable the search for diagnostic measures in other frequency bands in the future.

In the conducted experiment, with the known and correct technical condition of the gears, these components were not the result of damage, but the location of the measuring point on another structural element of the housing (point 4 on the housing close to the 3rd-stage transmission—planetary gear; and point 5 on the top cover of the housing).

The results of the multidimensional data interpretation of vibration signals of system condition monitoring of the gear transmission operating under difficult conditions for STFT distributions for the three selected measurement points are depicted in Figure 20. These results made it possible to evaluate the dynamic stability of the machine by comparing the distribution of the STFT amplitude values for a specific frequency band over the entire duration of the measurements. Such an analysis method is shown in Figure 20, which indicates an exemplary observed frequency band with broken-line cylinders. Such an analysis and observation was not possible on the basis of the FFT spectra, which prevented the correct identification of the operating condition of the transmission.

The STFT distributions of vibration signals at various measurement points clearly showed quantitative and qualitative differences in vibrations, as well as the presence of local extremes occurring only for a specific time and frequency, which will be masked and invisible in waveforms and FFT spectrums.

## 6. Conclusions

The results of studies on vibration signals of a three-stage transmission with a bevel–cylindrical–planetary configuration installed in an experimental scraper conveyor showed some important factors, and allowed us to formulate interesting conclusions. The developed processing algorithm allowed us to obtain a multidimensional data interpretation of vibration signals of system condition monitoring of a transmission operating under difficult conditions. It allowed us to obtain a preliminary assessment concerning the possibility of using vibration signals in the diagnostics and evaluation of the technical condition of the power transmission system in a scraper conveyor’s driving transmission. In the case of such complex machines operating in difficult conditions in complex mechanical systems, there is often a phenomenon of overlapping symptoms of the condition of individual machine elements and disorganization of components resulting from damage or wear. Thus, the separation and classification of symptoms on the basis of vibration measures is a very big challenge. Due to such a difficult research issue, an active diagnostic experiment was conducted, the aim of which was to compare the identification quality of the technical condition of the gear transmission, depending on the mounting location of the vibration sensor.

Some conclusions are as follows:(1)To analyze and evaluate the information content in waveform vibration signals recorded at various measuring points, quantitative estimators were determined. The estimators were selected to represent the energy content of the signal, the distribution of the maximum values, and measures of dissipation and dispersion.(2)The largest range of extreme values was obtained for the coefficient of variation (CoV)—the percentage range was 418% (for the X-axis vibration); the smallest was obtained for RMS—74% (for the X-axis vibration). Each range of percentage values was much too large, which confirmed the large influence of the location of the measuring point on the obtained results of the vibration estimators.(3)The data shown in Figure 18 confirmed the significant dispersion of values. It is visible in the figure that, assuming certain SHM control thresholds, there was a high risk of exceeding their values and of an incorrect alarm condition that would result only from the incorrect selection of the sensor location (not damage).(4)FFT enabled the identification of components related to the dynamics of the gear transmission operation. This allowed for the identification of parameters of kinematic nodes and other dynamic phenomena, including changes in machine parameters resulting from lab damage. In the case of overlapping of other components in the spectra, the diagnostic result may have been difficult or erroneous.(5)A strong correlation of the characteristic frequencies was clearly visible (Figure 19); however, significant differences could be observed when comparing the amplitude values of the dominant FFT components. In the cases of points 4 and 5 in the FFT spectra, local extremes in the frequency bands were also visible, but were negligibly small for points 1–3.(6)For the multidimensional analysis of the signal, the time distribution of individual frequency components was clearly visible, which was not always constant. Such information about changes in the amplitudes of specific frequency components over time may also be essential in SHM systems, especially in the diagnostic aspect of complex machines operating in heterogeneous and difficult conditions, such as mining conveyors.(7)The STFT distributions of vibration signals at various measurement points clearly showed quantitative and qualitative differences in vibrations, as well as the presence of local extremes occurring only for a specific time and frequency, which were masked and invisible in waveforms and FFT spectrums.

The developed method of comprehensive analysis of vibration signals enabled full identification and selection of significant signal components in the amplitude, frequency, and time–frequency dimensions. The related measurements allowed a preliminary analysis leading to the identification of appropriate bands and lines in the frequency domain, relative to the driving transmission’s design and ratio (frequency of revolution, and meshing or cooperation between the driving chain and the driving gear), all aimed at selecting specific frequency ranges and preliminarily adopting them for the sake of potential diagnostics of vibrations.

An additional advantage of the diagnostic algorithm developed in this way was in the case of observing short-term and atypical STFT components or discontinuities; e.g., as a sudden reduction of the elementary frequency amplitude, identifying the time of occurrence of this phenomenon, and returning to the detailed observation of a specific waveform of the signal time period.

The developed method enabled the analysis of the diagnostic capacity of vibration signals and the precise selection of measurement points where system condition monitoring or SHM sensors will be intentionally installed. 

In future studies, testing will be expanded to verify the results in comparison to other independent monitoring systems in different operating scenarios.

## Figures and Tables

**Figure 1 sensors-21-07808-f001:**
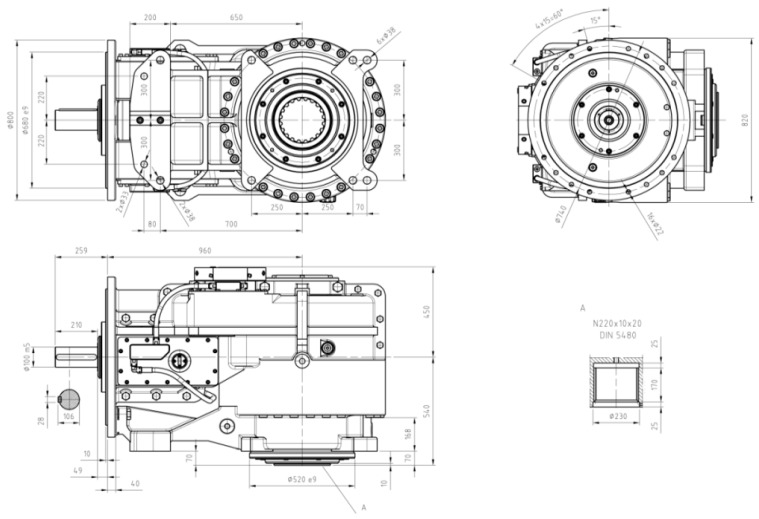
Service dimensions of the type KPL-25 bevel gear transmission.

**Figure 2 sensors-21-07808-f002:**
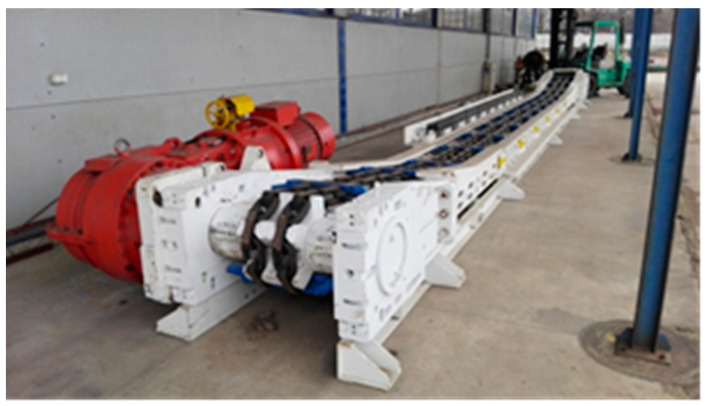
The test transmission installed in the scraper conveyor.

**Figure 3 sensors-21-07808-f003:**
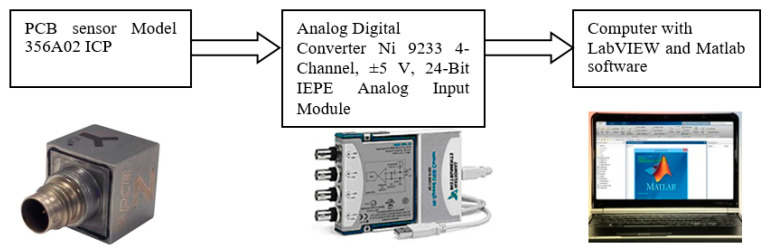
Measurement system.

**Figure 4 sensors-21-07808-f004:**
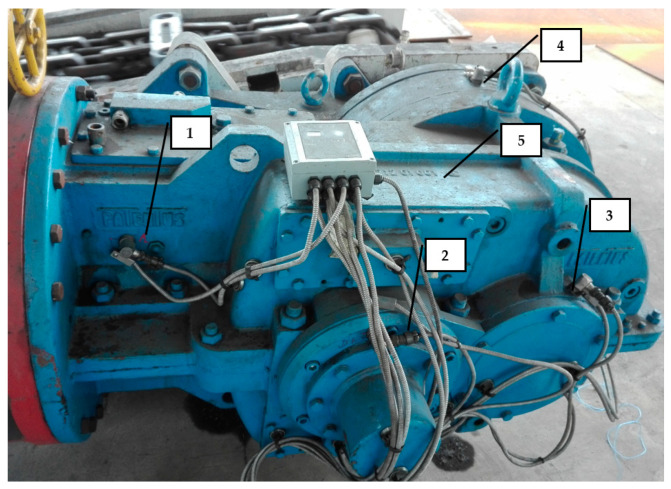
The sensor locations.

**Figure 5 sensors-21-07808-f005:**
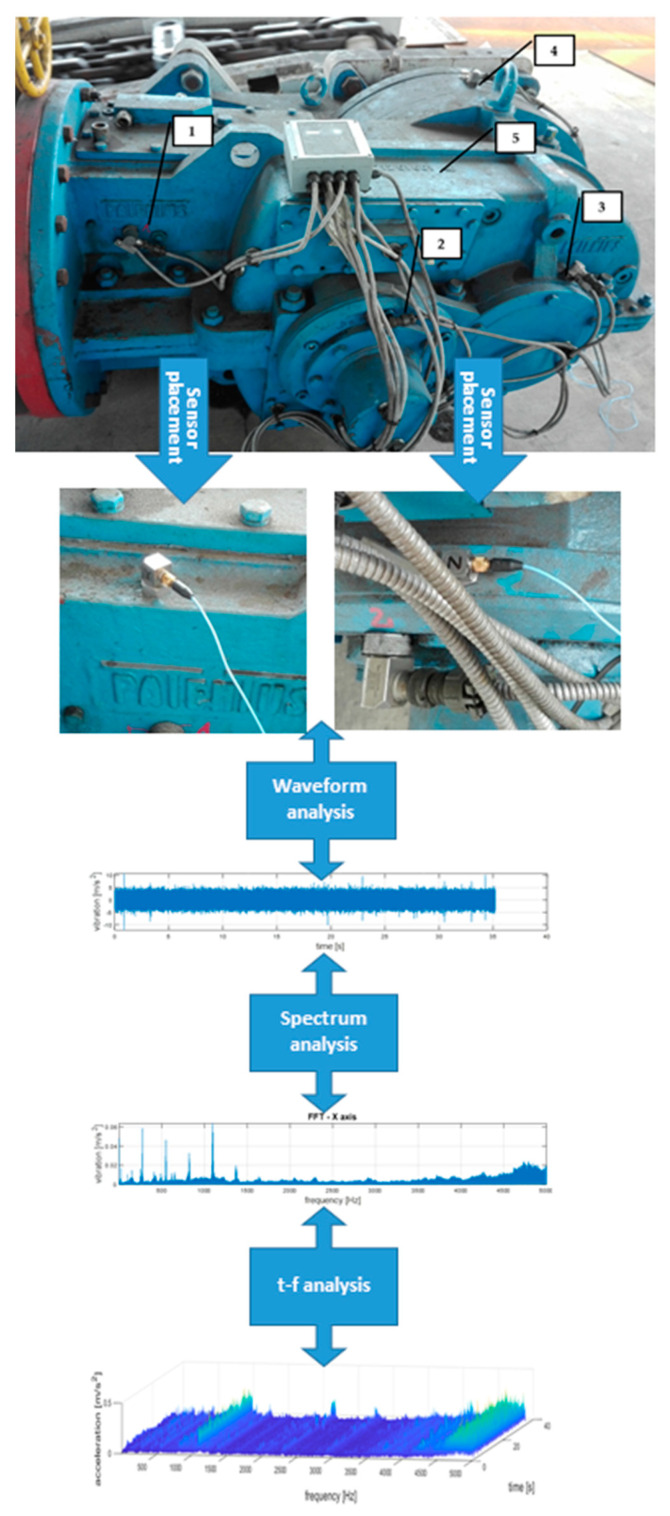
Research design diagram.

**Figure 6 sensors-21-07808-f006:**
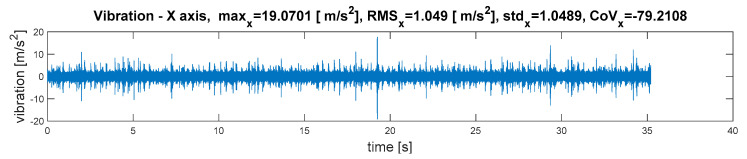
Waveform of vibration and time-varying quantity estimators.

**Figure 7 sensors-21-07808-f007:**
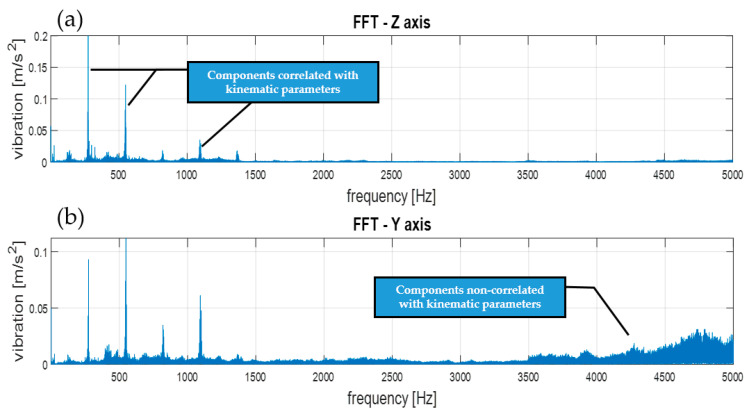
FFT spectra of vibration and main components correlated with the dynamics of the machine operation: (**a**)—spectrum of Z axis vibration, (**b**)—spectrum of Y axis vibration.

**Figure 8 sensors-21-07808-f008:**
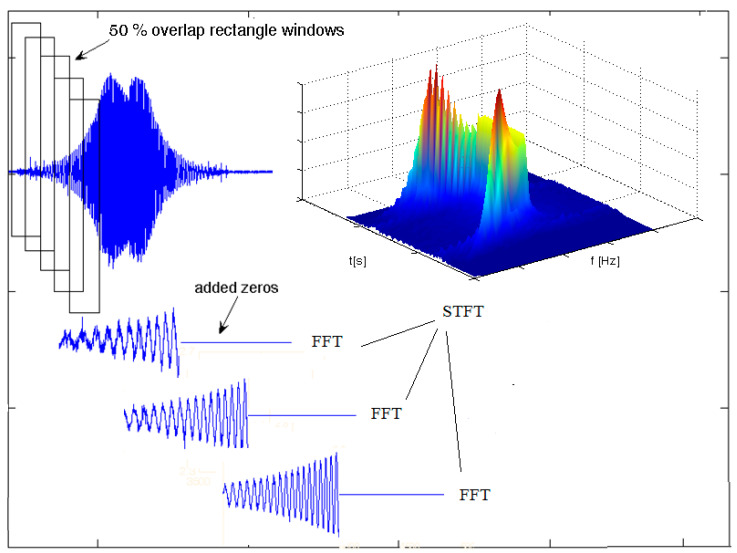
STFT transformation procedure with an overlap and added zeros.

**Figure 9 sensors-21-07808-f009:**
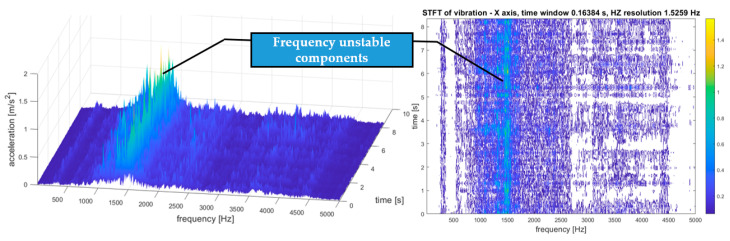
STFT distribution of the vibration signal of the tested transmission gear with view of frequency unstable components.

**Figure 10 sensors-21-07808-f010:**
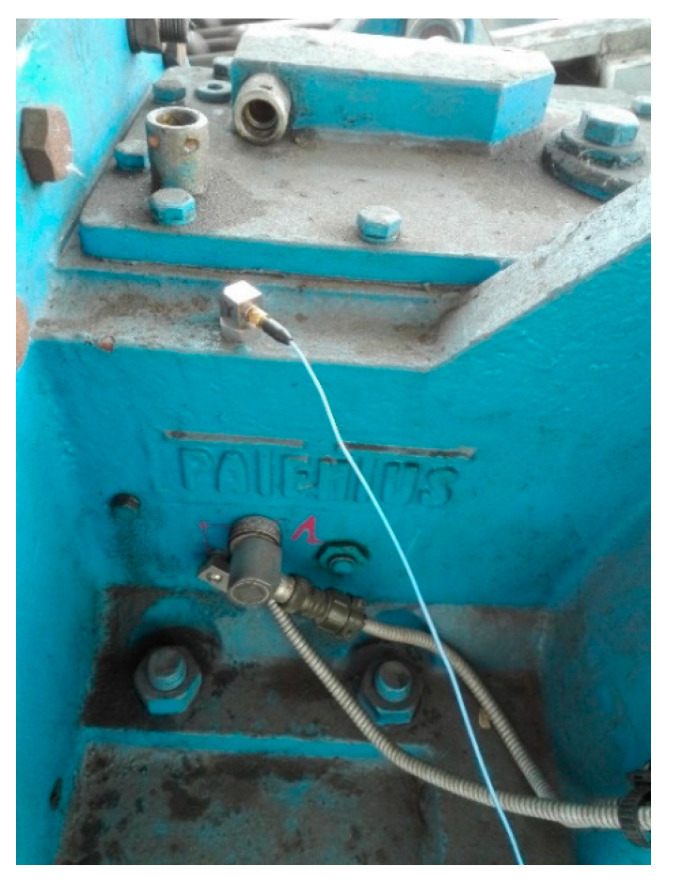
Measurement point 1—housing of 1st-stage transmission—bevel gear.

**Figure 11 sensors-21-07808-f011:**
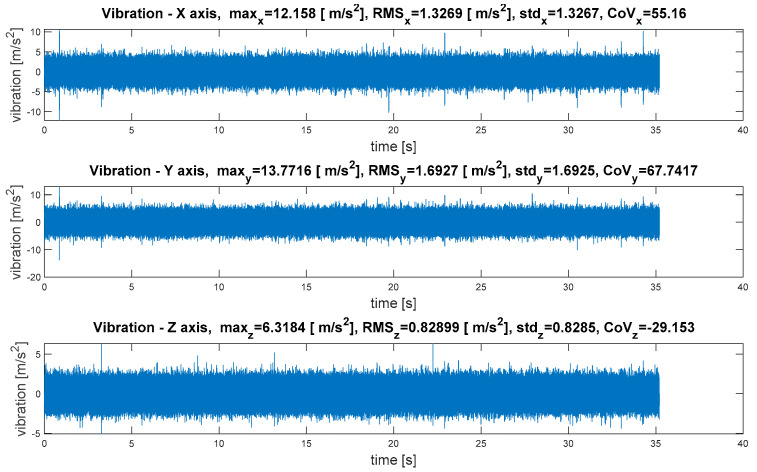
Measurement point 1—waveforms of vibrations in three orthogonal axes.

**Figure 12 sensors-21-07808-f012:**
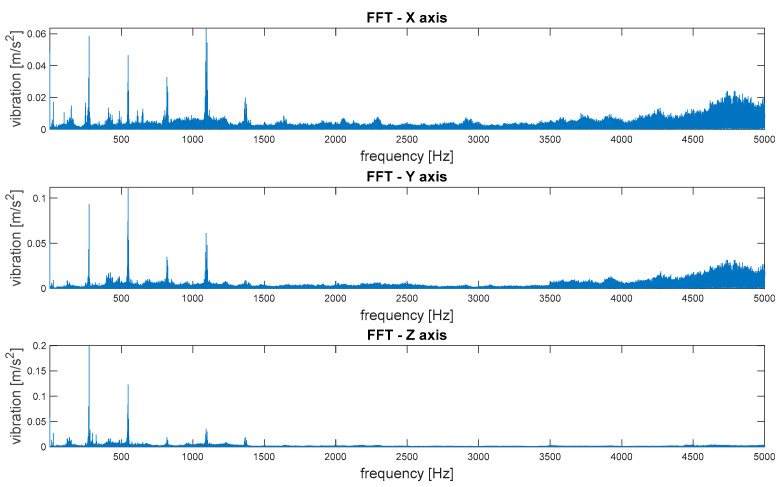
Measurement point 1—spectra of vibrations in three orthogonal axes.

**Figure 13 sensors-21-07808-f013:**
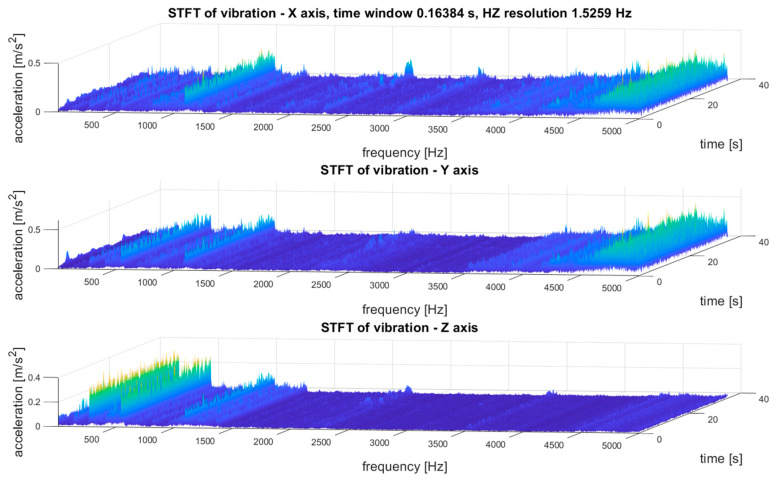
Measurement point 1—STFT of vibrations in three orthogonal axes.

**Figure 14 sensors-21-07808-f014:**
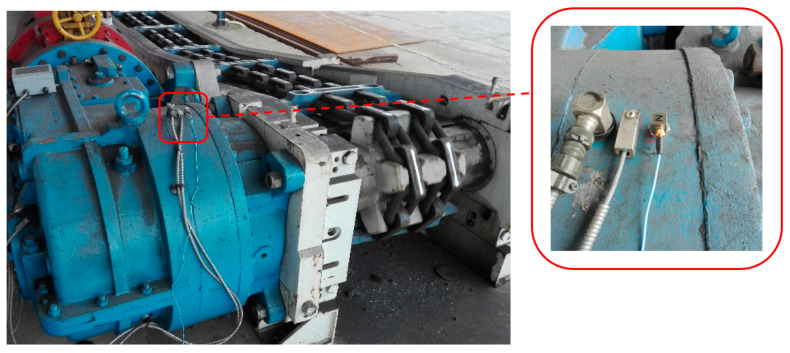
Measurement point 4—housing of 3rd-stage transmission—planetary gear.

**Figure 15 sensors-21-07808-f015:**
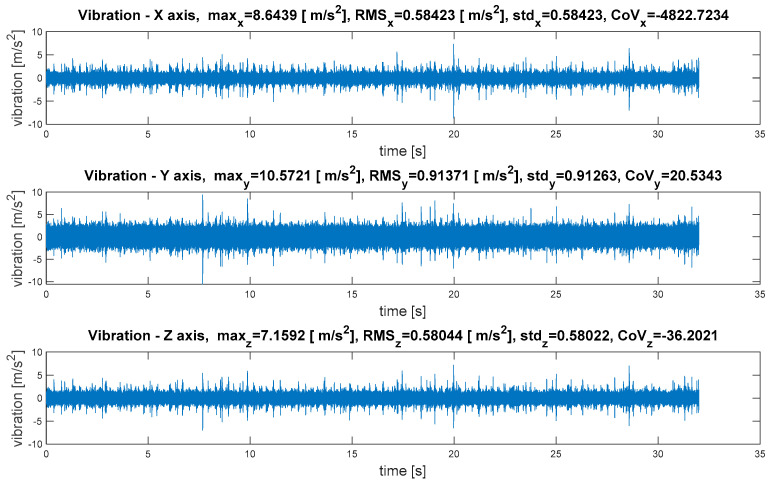
Measurement point 4—waveforms of vibrations in three orthogonal axes.

**Figure 16 sensors-21-07808-f016:**
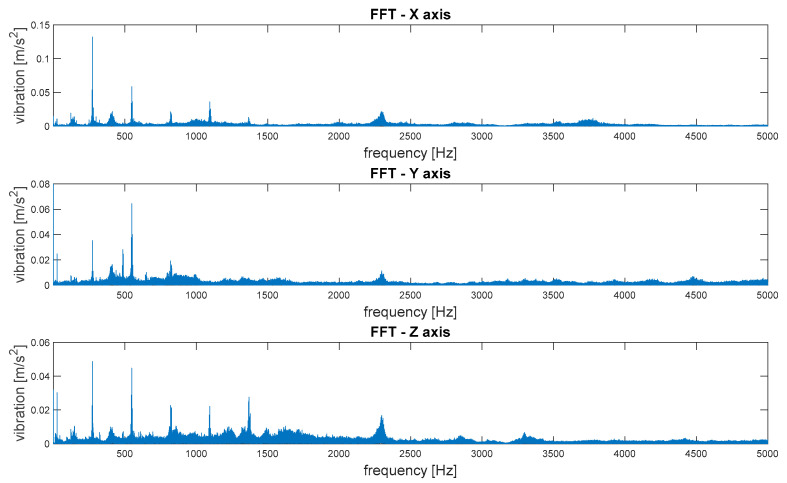
Measurement point 4—spectra of vibrations in three orthogonal axes.

**Figure 17 sensors-21-07808-f017:**
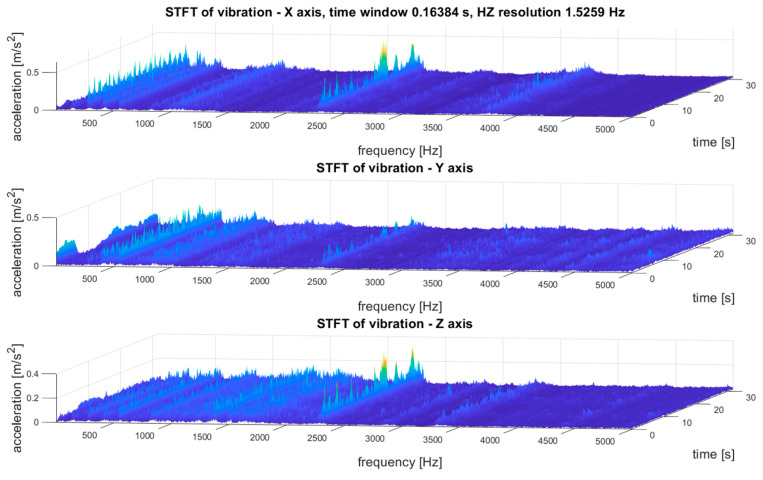
Measurement point 4—STFT of vibrations in three orthogonal axes.

**Figure 18 sensors-21-07808-f018:**
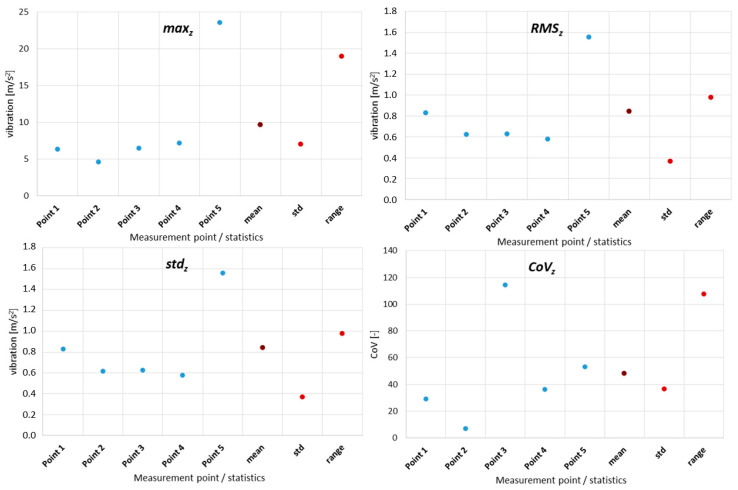
Distribution of quantity estimators of waveforms of vibrations (Z-axis).

**Figure 19 sensors-21-07808-f019:**
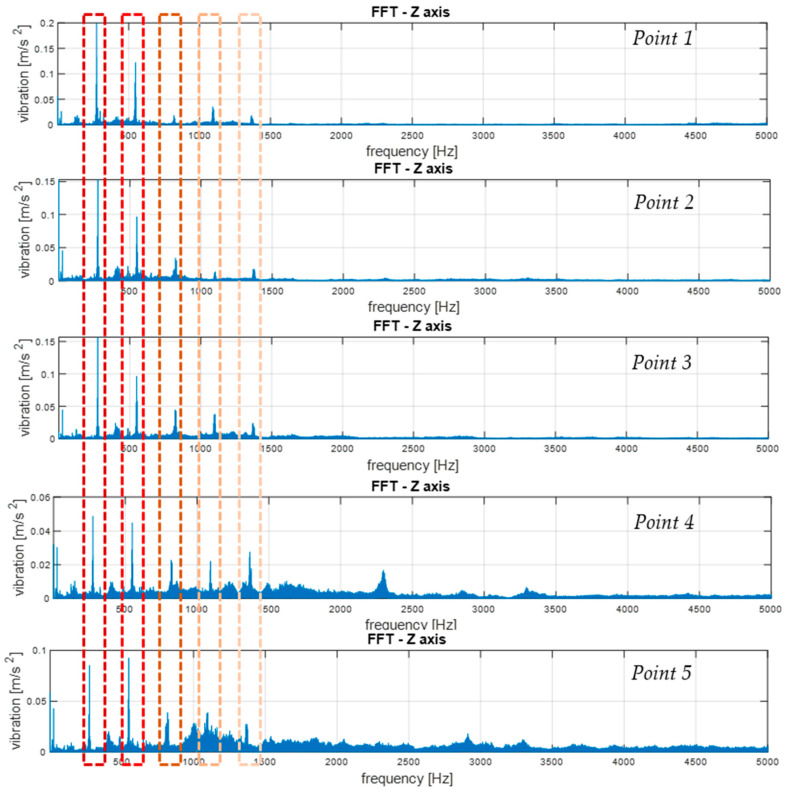
Comparison of the spectra of vibrations along the vertical axis (Z) determined for signals recorded at various measurement points.

**Figure 20 sensors-21-07808-f020:**
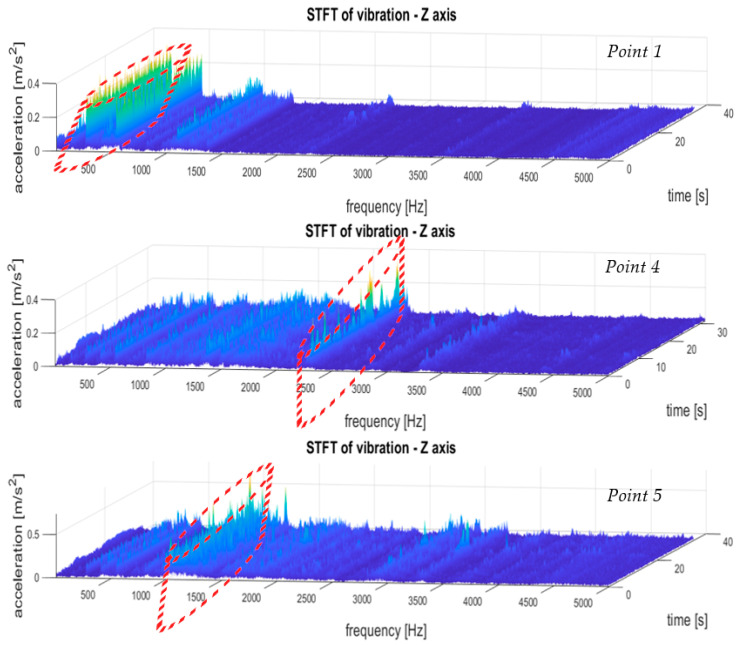
STFT distribution for three different measurement points.

**Table 1 sensors-21-07808-t001:** Kinematic parameters of the type KPL-25 bevel gear transmission.

Transmission Type:	Three-Stage Transmission of Bevel–Cylindrical–Planetary Configuration
Total ratio	39.326
Transmission’s input shaft torque	650 Nm
Transmission’s input shaft rotational speed	1470 rpm
Transmission’s output shaft torque	25,550 Nm
Transmission’s output shaft rotational speed	37.38 rpm

**Table 2 sensors-21-07808-t002:** General parameters of the measurement system.

Parameters:	Value
PCB 356A02 sensitivity (±10%)	10 mV/g
PCB 356A02 measurement range	±500 g pk
PCB 356A02 nonlinearity (400 g)	≤1%
PCB 356A02 temperature range	−54 to 121 °C
Ni 9233 ADC resolution	24 bits
Ni 9233 number of channels	4 analog inputs
Ni 9233 type of ADC	Delta-Sigma
Ni 9233 sampling mode	Simultaneous

**Table 3 sensors-21-07808-t003:** Quantity estimators of waveforms of vibrations in three axes for five different measurement points.

	*max_x_*	*RMS_x_*	*std_x_*	*CoV_x_*	*max_y_*	*RMS_y_*	*std_y_*	*CoV_y_*	*max_z_*	*RMS_z_*	*std_z_*	*CoV_z_*
*Point 1*	12.158	1.327	1.327	55.160	13.772	1.693	1.693	67.742	6.318	0.829	0.829	29.153
*Point 2*	7.080	0.834	0.834	44.979	11.861	1.202	1.202	52.506	4.603	0.625	0.618	7.029
*Point 3*	14.034	1.233	1.233	715.034	11.195	0.803	0.803	24.132	6.463	0.627	0.627	114.519
*Point 4*	8.644	0.584	0.584	4822.723	10.572	0.914	0.913	20.534	7.159	0.580	0.580	36.202
*Point 5*	19.070	1.049	1.049	79.211	19.795	0.988	0.988	140.693	23.566	1.557	1.557	53.122
*mean*	*12.197*	*1.005*	*1.005*	*1143.421*	*13.439*	*1.120*	*1.119*	*61.121*	*9.622*	*0.844*	*0.842*	*48.005*
*std*	*4.229*	*0.270*	*0.270*	*1857.106*	*3.354*	*0.315*	*0.315*	*43.509*	*7.023*	*0.367*	*0.368*	*36.398*
*range*	*11.991*	*0.743*	*0.742*	*4777.744*	*9.223*	*0.889*	*0.890*	*120.158*	*18.963*	*0.977*	*0.977*	*107.490*
*variance*	*17.885*	*0.073*	*0.073*	*3,448,842.351*	*11.250*	*0.099*	*0.099*	*1893.039*	*49.317*	*0.135*	*0.135*	*1324.810*

## Data Availability

Data supporting reported results can be found on: https://polslpl-my.sharepoint.com/:f:/g/personal/rburdzik_polsl_pl/EnDBwBXp1Y5OpE31wggkbl0BkVjSe5ejM95D6BSrQ-G9EQ?e=nGpEAU (accessed on 21 November 2021).

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
