# Peer review of "Multidimensional Data Interpretation of Vibration Signals Registered in Different Locations for System Condition Monitoring of a Three-Stage Gear Transmission Operating under Difficult Conditions"

_sensors, 2021, doi:10.3390/s21237808_

Round 1

Reviewer 1 Report

The paper analyses a system condition monitoring involving a  3-stage transmission of bevel-cylindrical-planetary configuration installed in a scraper conveyor.

The paper is written in a professional way. The mathematical and signal processing tools are not new, but the paper is still of interest with particular focus on the experimental setup.

The review of the literature is adequate, the description of the sensor is detailed, and the analysis of the results is well detailed.

Soem aspects for the authors to consider:

  • may be there is some excess of photos and one or two can be deleted.
  • some extra details about the real-time instrumentation, signal processing and computational system would benefit the paper.
  • Some discussion about other alternative techniques would also be interesting
  • Finally, deeper discussion the effects of noise and its treatment would be of relevance.

Reviewer 2 Report

The authors proposed a multidimensional data interpretation method to detect the potential fault scenarios of 3-stage gear transmissions. The time-frequency characteristics of the monitored components are obtained through time-frequency transformation. This reviewer has the following three questions that need further explanation from the authors.

-1-. Why did the authors choose 50kHz as the sampling frequency of the target component? Because this sampling frequency is a bit high, the scale of the collected data is too large, causing computational burden.

-2-. Does this research consider the impact of noise? Especially when dealing with the frequency characteristics of multi-dimensional signals, the influence of noise should be included.

-3-. When carrying out time-frequency conversion, such as using the STFT method, different time windows often have an impact on the time-frequency characteristics. Another factor is that when the speed of the gear transmission changes, the time-frequency characteristics will also change. Have the authors considered the time-frequency analysis in the case of variable speed?
